# Evaluation of Factors Influencing the Inclusion of Indigenous Plants for Food Security among Rural Households in the North West Province of South Africa

**Abiodun Olusola Omotayo** [1] and **Adeyemi Oladapo Aremu** [1,2,*]

1   Food Security and Safety Niche Area, Faculty of Natural and Agricultural Sciences, North-West University, Private Bag X2046, Mmabatho 2790, North West Province, South Africa; 25301284@nwu.ac.za

2   Indigenous Knowledge Systems Centre, Faculty of Natural and Agricultural Sciences, North-West University, Private Bag X2046, Mmabatho 2790, North West Province, South Africa

*   Correspondence: Oladapo.Aremu@nwu.ac.za; Tel.: +27-18-389-2573

**Abstract:** Underutilised indigenous plants can support and strengthen the existing food system, as they are considered as socio-economically and environmentally appropriate. These plants generally adapt to marginal conditions, which is essential for a resilient agriculture and sustainable food systems. The current study relied on food security and indigenous plants data collected from some selected rural households from the North West Province of South Africa. The utilised data were collected through a multi-stage sampling technique with the aid of a pre-tested semi-structured questionnaire, while descriptive methods Foster–Greer–Thorbecke (FGT) and binary logistic regression were used for data analysis. The models produced a good fit for the data, and the computed F-value was statistically significant ($p < 0.01$). The study examined socio-economic and food security status based on the knowledge and the perception of indigenous plants by the households. The incidence of food insecurity ($\theta_0$) was 0.4060, indicating that 40.6% of the participants were food insecure while 59.4% were food secured. Binary logistic regression results indicate that factors such as age, gender, educational attainment, inclusion of indigenous plants in diet, food expenditure, and access in the study area impacted results. It was also evident that the participants had considerable knowledge of indigenous plants. However, these indigenous plants were not cultivated or included in the diet by the majority of the participants. The formulation of appropriate holistic policies that support the incorporation of the indigenous plants into the food system is recommended.

**Keywords:** binary logistic regression; food system transformation; food shortage; undervalued plants; sustainability

## 1. Introduction

The global nutrition report (GNR) 2020 emphasized the increasing role of nutrition in health and well-being, even in the face of diverse challenges confronting the global food systems [1,2]. The need to provide the increasing global population with healthy diets from a sustainable food system cannot be overemphasized [3]. The United Nations (UN) World Food Programme warned that an estimated 265 million people could face food insecurity by the end of 2020 due to the impact of COVID-19 [4–6]. Based on recent projections, the economies in Africa will be severely affected, as almost 9% economic decline has been predicted [7]. As applicable with most developing countries, South Africa continues to experience food insecurity [8–10]. Poverty-stricken households often lack money to buy food and

are unable to produce their own food and are typically characterized by few low income-earners and many dependents particularly vulnerable to economic shocks (http://www.statssa.gov.za).

Recently, the importance of indigenous plants in mitigating food and nutrition insecurity, particularly in sub-Saharan Africa, has been widely recognized [11–15]. Indigenous plants have multiple uses within society and most notably contribute towards the diversification of food intake to enhance food and nutrition security. However, research suggests that the benefits and the value of indigenous plants within the South African context have not been fully understood and synthesized [16]. In addition, factors that determine the inclusion of these indigenous plants in the diets among households in South Africa remain unknown. Interestingly, the potential value of the indigenous plants to the South African food system could be harnessed if their benefits can be explored. Studies have suggested that the use of indigenous plants is currently low due to the non-availability of these plants in modern markets/shops and the lack of investment in research development [14,17]. Thus, indigenous plants have remained less competitive relative to the existing exotic crop varieties [18,19].

In South Africa, food insecurity needs urgent attention, especially from a local perspective which will be sustainable overtime. As a result, the aim of this study was to analyse the factors influencing the inclusion of indigenous plants for food security among rural households in the North West Province of South Africa. The factors influencing the inclusion of indigenous plants were evaluated to establish and understand policy instruments for communicating a sustainable food system and food security strategies. Currently, there is a scarcity of studies in South Africa that have modelled indigenous plants potential for food security using a robust model such as binary logistic regression. These identified gaps need to be filled through an interdisciplinary research approach for proper design and implementation of indigenous plant-inclusive food policies for a more sustainable food system. For the current study, the null hypothesis is that the socio-economic characteristics of the participants do not have a relationship with their food security status, while the alternative hypothesis indicates that the socio-economic characteristics can deliver the food security status of the participants. To achieve the aim of the study, the following three research questions guided the research.

(1)     What is the socio-economic and food security status of the participants?
(2)     What are the factors influencing the inclusion of indigenous plant-diet for food security?
(3)     How knowledgeable are the households on the potential of indigenous plants for food security?

## 2. Review Synthesis from Literature on the Food System and Potential of Indigenous Plants

A sustainable and healthy food system can deliver food and nutrition security for all in a way that is economically, socially, and environmentally sound [20]. Underutilised indigenous plants can support and strengthen the existing diet, especially that of the rural communities of South Africa as well as the food system. Several underutilised indigenous plants are known to be rich in nutrients and can adapt to marginal conditions. This translates in their potential to champion sustainable food systems if incorporated in the diet for food security [21].

### 2.1. Overview of Indigenous Plants, Food Security, Diet, and Current Food System in South Africa

Globally, the increasing food shortage and food insecurity have been aggravated by the COVID-19 pandemic [22,23]. In South Africa, there is an urgent need for the empowerment of the historically marginalized rural communities, which can be facilitated by the inclusion of underutilised indigenous plants in small holder farming [24,25]. The current situation of the food system remains a reflection of the apartheid regime [20]. The current diet pattern relies on the exotic food varieties such as rice, wheat, potatoes, apple, plum, and spinach, which are relatively expensive and unaffordable for many rural dwellers. This has resulted in an increased burden of food shortage, hunger, and malnutrition on the marginalized citizens. Although the current food system has the capacity to feed the

South African population [20,26], food remains inaccessible to approximately 26% of the population [27]. This highlights the flaw in the existing food system and South Africa's food-nutrition blueprint.

Evidence abounds that food production of commercial crops such as maize, sunflower, and apple is increasing, but this is not enough to meet the needs of the increasing population in South Africa [20]. Food production has to increase geometrically to mitigate this challenge [3,28]. Agriculture in South Africa is the main source of livelihood for many poor rural households, however, it is operated under marginal conditions and often cannot sustain households [29–31]. Therefore, the inclusion of underutilised indigenous plants in the diets could be an alternative to bridge the food and nutrition insecurity gap. Several underutilised indigenous plant are nutrient-dense with good adaptability to marginal conditions and are more likely to be a sustainable and nutritious source of food for the rural communities [14,32].

### 2.2. Agricultural Food Policy, Agro-Biodiversity, and South Africa Food System

The agricultural policy priorities are reflected in the National Development Plan (NDP, 2030), which prioritizes the commercialization of crops which are in line with the dominant food system [33]. The National Food and Nutrition Security Policy has similar priorities, as it speaks about efficient agricultural production and is largely silent about the cultivation of underutilised plants [34–36]. However, these policies advocate for the consumption of the underutilised indigenous and traditional crops [20,37,38]. The current South African food policies reflect a developmental agenda and aim to develop small farmers but do not fully consider their limitations and opportunities for inclusion into the dominant food systems [39].

Indigenous plants remain poorly explored and are often excluded from policy and strategic documents. The recognition of the importance of indigenous plants for food security could be used to leverage their incorporation within policy implementation processes and to improve access to dietary diversity among the previously disadvantaged population [40]. The concerns on the gradual loss of biodiversity and vulnerability to climate change have prompted a call to rethink the current configuration of the South African food system [20,41]. A focus on re-invigorating underutilised indigenous plants and bringing these to the market has been suggested as an entry point for improving diets and making them more sustainable [42].

Given the evidence presented above, in what ways can the paradigm shift be steered to support the South African vulnerable populations and improve their food security and socio-economic status? What type of evidence should be created to guide policy makers in supporting the marginalized rural communities? The quality of the food for the vulnerable populations can be improved through the incorporation of the indigenous plants into the South African food system. To create an enabling environment for change at the local level, a multi-disciplinary research approach combining indigenous plants and econometrics with other scientific parameters is imperative to inform policy makers of the sustainable food landscape with the inclusion of indigenous plants in the food system.

## 3. Materials and Methods

### 3.1. Study Area

The study was conducted in the North West Province, South Africa. The province lies in the north of South Africa on the Botswana border, with the Kalahari Desert to the west, the Gauteng province to the east, and the Free State to the south [43]. The North West Province was established in 1994, making up 8.7% of South Africa's land area (106,512 km$^2$). The landscape is demarcated by Magaliesberg Mountain in the northeast, which extends to about 130 km from Pretoria to Rustenburg, while the Vaal River forms the province's southern border [44]. Based on the 2011 census, the population of North West Province is 3,509,953, with 65% of the population located in rural areas [45]. The climate in the province varies considerably, with the areas in the east being much wetter than those in the west. The province is dominated by a flat savanna and grassland landscape, which is home to rich biodiversity

and agriculture, with hills and ridges dividing up this landscape [46]. Mahikeng is the administrative capital, while the majority (80%) of the economic activities in the province are concentrated in the southern region between Potchefstroom and Klerksdorp as well as in the Rustenburg and the eastern region [47,48]. Mining is the major contributor to the economy, followed by agricultural activities which include a range of commercial crop farming, including maize and wheat, livestock farming, game farming, and subsistence farming.

The North West Province consists of four district municipal councils, namely: Ngaka Modiri Molema, Bojanala Platinum, Dr. Ruth Segomotsi Mompati, and Dr. Kenneth Kaunda district municipality (Figure 1). These district municipalities are in turn divided into 18 local municipalities. Agriculture is an important activity in the North West Province because of its contribution to food security in the country. Even though 43.9% of the land in the province is classified as arable, only 20.9% is currently cultivated [48,49]. Socio-economics and food insecurity in the rural North West province were key drivers for this study. In addition, the North West Province has a rich biological and ethnic diversity in South Africa, thereby providing an exciting prospect for scientific research and innovations that could be of major cultural and commercial significance nationally and globally [15,49].

### 3.2. Sampling Methods and Sample Size

Multi-stage sampling procedure was used in this study. Prior to the commencement of the survey, the enumerators were properly trained on the requirements of the survey, and a pre-testing of the questionnaire was undertaken among a few rural households. Although the questionnaire was designed in English language, interviews were conducted for the majority of the participants in their local languages, which were mainly Setswana and Northern Sotho. At the first stage of the sampling, the four district municipalities (Ngaka Modiri Molema, Dr Kenneth Kaunda, Bojanala Platinum, and Dr Ruth Segomotsi Mompati District Municipality) were selected. The second stage involved choosing three communities from each district municipality, making a total of 12 communities. The selection of the 12 communities was based on their rural nature, small holder agricultural practices, cultures, as well as the low socio-economic status of the residents. The last and final stage of sampling was the selection of rural household heads, which was simplified by the assistance of residence extension officers. Forty-five questionnaires were administered in each of the four district municipalities, which made a total of 180 questionnaires. After thorough screening, 133 properly questionnaires were utilized in this study. This was made up of 33–34 properly filled questionnaires from each of the four district municipalities of the North West Province. Details of the number of responses from each of the selected villages are depicted in Figure 2.

### 3.3. Research Instrument, Validity, and Reliability

A semi-structured questionnaire (prepared in English language) was used for data collection. As listed in Table 1, we used an inventory of 21 indigenous plants (grains, vegetables, and fruits) recognised as important and popular based on information from different sources [12,14,50] and the South African government agency [51]. A photo album of these 21 selected plants was compiled and used as visual aid for ease of identification by the participating households (an average of 5 members in this study). The pictures were obtained from the aforementioned literature and reliable websites. The botanical names were verified using the Plant List (http://www.theplantlist.org/). The questionnaires, which were completed anonymously, were divided into two sections: (i) socio-economic characteristics and (ii) knowledge and consumption of indigenous plants. Prior to data collection, the questionnaires were tested and validated during the pilot study. The validity was done by pre-testing the data collection tool with a sample of 20 rural households, which were randomly selected and interviewed. A few questions that were found invalid, overlapping, or unnecessary during the pre-testing process were deleted, while the ambiguous ones were modified to ensure clarity. A half technique was used to determine the reliability of the instrument. A reliability coefficient of r = 0.80

was obtained, which was considered to be good for the instrument used. Therefore, the administered questionnaires were reliable, consistent, and accurate in response to the objectives of the study.

**Table 1.** Scientific name and plant families for the selected indigenous vegetables, grains, and fruits utilised in the study.

| Indigenous Vegetables and Grains | Indigenous Fruits |
| --- | --- |
| *Amaranthus* sp.—Amaranthaceae | *Annona senegalensis* Pers.—Annonaceae |
| *Cajanus cajan* (L.) Millsp.—Fabaceae | *Carissa macrocarpa* (Eckl.) A.DC.—Apocynaceae |
| *Cleome gynandra* L.—Cleomaceae | *Diospyros lycioides* Desf.—Ebenaceae |
| *Colocasia esculenta* (L.) Schott—Araceae | *Diospyros simii* (Kuntze) De Winter.—Ebenaceae |
| *Glycine max* (L.) Merr.—Fabaceae | *Dovyalis caffra* (Hook.f. & Harv.) Sim—Salicaceae |
| *Lagenaria siceraria* (Mol.) Standl.—Cucurbitaceae | *Dovyalis zeyheri* (Sond.) Warb.–Salicaceae |
| *Manihot esculenta* Crantz—Euphorbiaceae | *Mimusops zeyheri* Sond—Sapotaceae |
| *Tetragonia decumbens Mill.*—Aizoaceae | *Parinari curatellifolia* Planch. ex Benth.—Chrysobalanaceae |
| *Sorghum bicolor* (L.) Moench—Poaceae | *Sclerocarya birrea* (A.Rich.) Hochst.—Anacardiaceae |
| *Tylosema esculentum* (Burch.) A.Schreib.—Fabaceae | *Strychnos spinosa* Lam.—Loganiaceae |
| | *Vangueria infausta* Burch.—Rubiaceae |

## 3.4. Analytical Framework and Estimation Techniques

Two main analytical techniques (listed in Sections 3.4.1 and 3.4.2) were used for the current study. The first was the Foster–Greer–Thorbecke (FGT) which was used to assess the food security status of the household head. Additionally, two thirds of the mean per capita household food expenditure (MPCHFE) was used for the determination of food security line. The second analytical tool was the binary logistic regression model used in identifying the factors influencing the indigenous plant's inclusion for food security.

### 3.4.1. Foster–Greer–Thorbecke (FGT) Food Security Analysis

Foster–Greer–Thorbecke (FGT) class of decomposable poverty measure was adopted to highlight the various food security statuses of the households. Household food security line was drawn as two thirds of the mean per capita household food expenditure (MPCHFE), and statuses of the households were derived either as food secure or food insecure. Households whose MPCHFE was above the line were food secure, while those below were food insecure. Measures of food insecurity incidence ($\theta_0$), depth ($\theta_1$), and severity ($\theta_2$) were also estimated. Similar to previous studies [52,53], the FGT measures were mathematically derived as:

$$P_\propto = \frac{1}{N}\left(\frac{z - y_i}{z}\right)^\propto 1(y_i \le z) \tag{1}$$

where $1(y_i \le z)$ denotes that food insecurity gap does not exist for households with mean per capita expenditure above the food security line. $\alpha$ = FGT food insecurity index which takes values 0, 1, 2 for $P_0$ = food insecurity headcount, $P_1$ = food insecurity depth, $P_2$ = food insecurity severity, respectively. It is also referred to as the elasticity of an individual's food insecurity with respect to the normalized gap $(z - y_i)$ such that a 1% increase in the insecurity gap of a food insecure person leads to an $\alpha$% increase in the individual's food insecurity level.

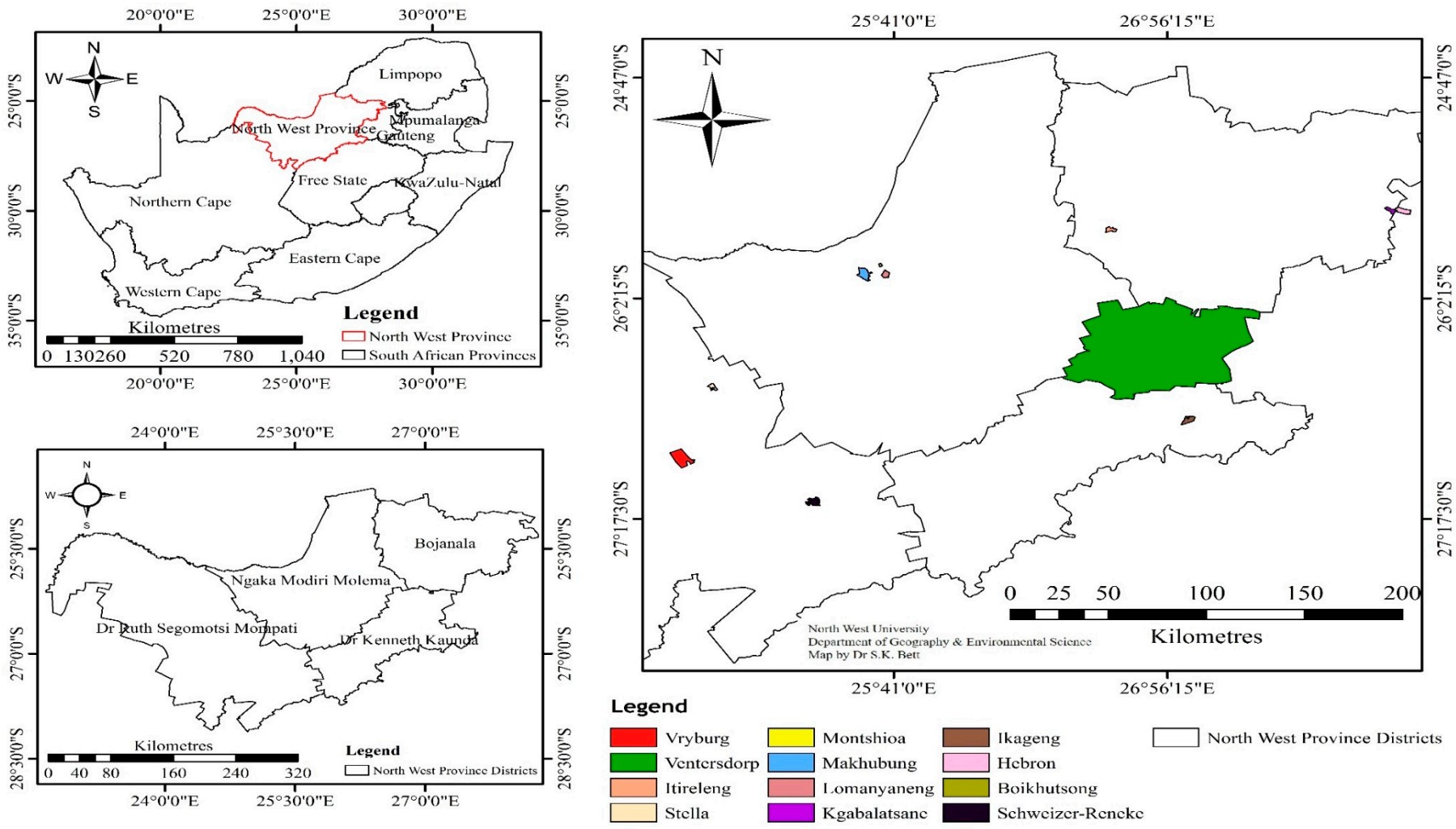

**Figure 1.** Location of the selected communities in North West Province of South Africa.

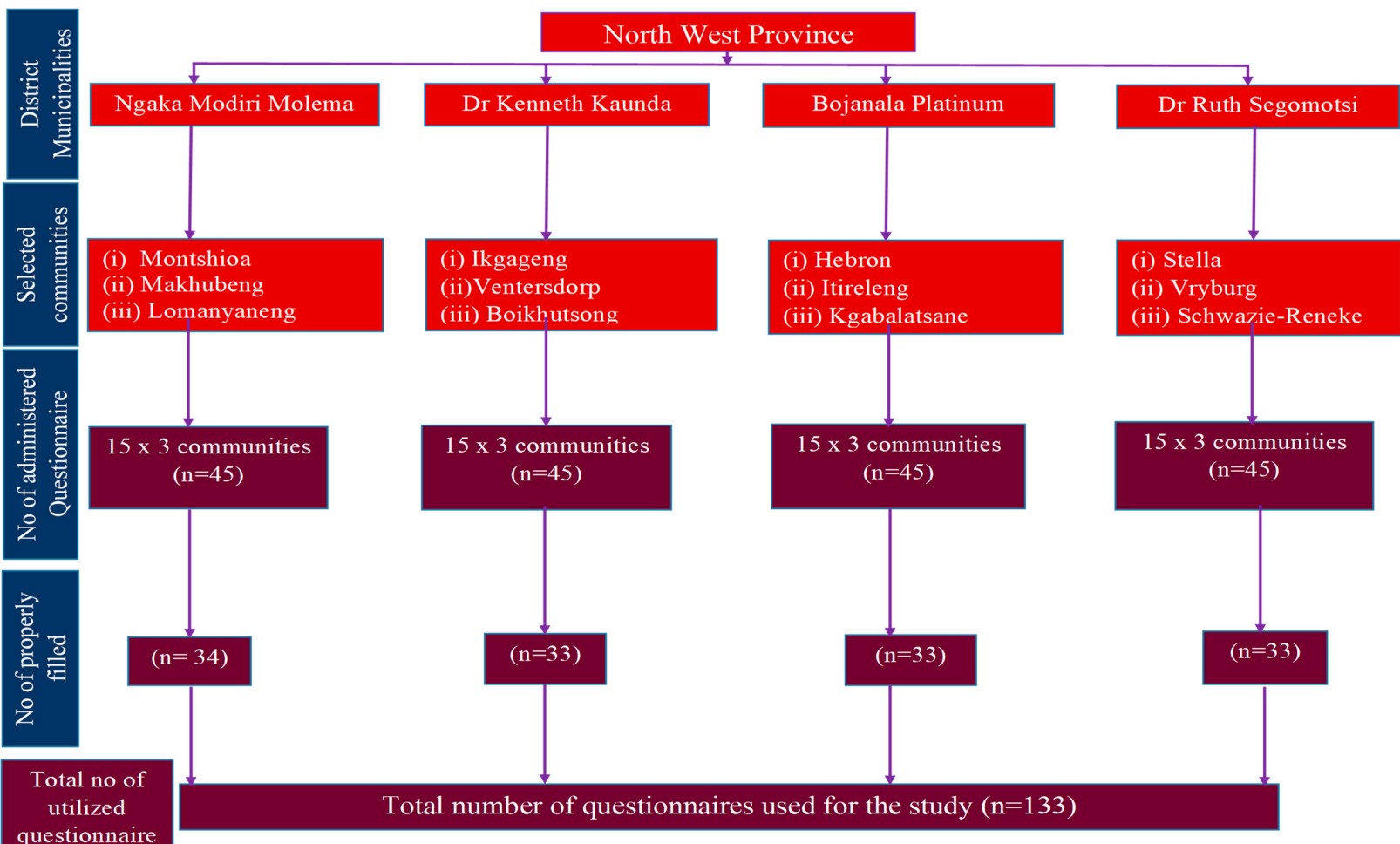

**Figure 2.** Framework used for sampling method and sample size for the study.

$N$ = total number of households $\beta_1$

$z$ = food security line

$P$ = number of households below the food security line

$y_i$ = the per-capita monthly food expenditure of the household in which individual $i^{th}$ lives.

### 3.4.2. Binary Logistic Regression Model for Determination of Factors Influencing the Inclusion of Indigenous Plants for Food Security

Binary logistic regression model was employed to determine the factors influencing the inclusion of indigenous plants for food security. The model is stated thus:

$$\text{Fi} = \beta_0 + \beta_1 X_1 + \beta_2 X_2 + \beta_3 X_3 + \beta_4 X_4 + \beta_5 X_5 \ldots + \beta_n X_n \tag{2}$$

Fi is the binary variable with value 1 if participants re-categorized food secured and 0 otherwise, where $\beta_2$ is the intercept (constant), and $\beta_1$, $\beta_2$, $\beta_3$, $\beta_4$, $\beta_5$ and $\beta_n$ are the regression coefficients of the predictor variables, $X_1$, $X_2$, $X_3$, $X_4$, $X_5$, and $X_n$. Binary logistic regression model is widely used to analyse data with dichotomous dependent variables. Hence, it was considered a suitable model to use for this objective because the dependent variable was dichotomous in nature. In addition, it was necessary to create dummy variables to use the selected socio-economics knowledge on indigenous plants, identification, and welfare variables. The independent variables used in the analysis are shown in Table 2.

**Table 2.** Factors influencing indigenous plants inclusion for household's food security.

| Variables | Description |
|---|---|
| **Dependent variable** | |
| Food security status | Dummy, 1 if yes, 0 if otherwise |
| **Independent variables (based on head of the household)** | |
| Age | Number of years (Continuous) |
| Gender | Dummy; 1 if head is male and 0 if otherwise |
| Marital status | Dummy; 1 if head is married, 0 otherwise |
| Educational attainment | Years of academic education (Continuous) |
| Religion | Dummy; 1 if head is a Christian, 0 otherwise |
| Income | Total value in Rands (Continuous) |
| Family size | Number of members of household (Continuous) |
| Number of working individuals | Number that works and have income (Continuous) |
| Occupation | 1 if civil servants, 0 otherwise |
| Inclusion of indigenous plants in diet | Dummy, 1 if yes, 0 if otherwise |
| Knowledge on indigenous plants | Dummy, 1 if yes, 0 if otherwise |
| Drought resistant nature of indigenous plants | Dummy, 1 if yes, 0 if otherwise |
| Households expenditure | Total value in Rands (Continuous) |
| Expenditure on indigenous plants | Total value in Rands (Continuous) |
| Accessibility to indigenous plants' market | Dummy, 1 if yes, 0 if otherwise |
| Implementing policies on indigenous plants | Dummy, 1 if yes, 0 if otherwise |
| Backyard cultivation of indigenous plants | Dummy, 1 if yes, 0 if otherwise |
| Mainstreaming indigenous plants into food system | Dummy, 1 if yes, 0 if otherwise |

## 4. Results and Discussion

### 4.1. Food Security Status of the Participants

In the current study, the food insecurity parameters used were $\theta_0$, $\theta_1$, and $\theta_2$, which mean food insecurity incidence (headcount), depth, and severity, respectively (Table 3). Food insecurity head count ($\theta_0$) represents the proportion of household below the food security line [53–56]. The current findings revealed a high degree of effectiveness by showing severity, depth, and incidence of food insecurity. The incidence of poverty ($\theta_0$) was 0.4060, indicating that 40.60% of the participants were food insecure, while 59.40% were food secured (Table 3). The value $\theta_1$ (poverty depth) among the rural participants was 0.1626, meaning that each household member would require 16.26% of the food insecurity line (R118.68) per day to be out of their present food insecurity status. The value $\theta_2$ (food insecurity severity) of the participants was 0.0869, indicating that the food insecurity severity of the participants was 8.69%. Additionally, an average core food insecure household would require about 8.69% of the food insecurity line to the households' food budget in order move out of their severe food insecurity status. From the findings, it could be inferred that there is existence of food insecurity among the rural households in the study area. This is in line with previous findings that food remains inaccessible to approximately 26% of the population [27] and confirms the assertions of existing literature [57–60].

**Table 3.** Food insecurity levels among the participating households.

| Food Insecurity Indices | Value |
| --- | --- |
| $\theta_0$ | 0.4060 |
| $\theta_1$ | 0.1626 |
| $\theta_2$ | 0.0869 |
| Mean per capita household food expenditure (MPCHHFE) | R177.13 |
| Food insecurity line (i.e., two thirds of MPCHHFE) | R118.68 |

### 4.2. Food Insecurity Indices of Participants Based on Socio-Economic Characteristics

Table 4 shows the socio-economic characteristics of the participants based on their food insecurity. Higher count ($\theta_0$) indicates higher incidence of food insecurity, higher $\theta_1$ implies higher depth of food insecurity, and higher $\theta_2$ values implies higher food insecurity severity situation. In the study, food insecurity incidence decreased with age. The highest food insecurity incidence (63.63%) was among households headed by individuals aged 21–30 years. Highest depth (34.75%) and severity (22.37%) were also among households headed by individuals within this age group. This is possibly due to the fact that most participants in this age category are young and inexperienced. As a result, these individuals probably earn lesser or no income, which could translate into food shortage and food insecurity among their households [61].

In the current study, incidence of food insecurity of 41.33% was higher among the female-headed households than the 40.35% found among their male counterparts. In addition, the female-headed households need 17.69% increase in daily per capita food expenditure in order to be food secure. This is against the 14.66% increase required by the male-headed households. This is in line with the existing findings that female-headed households tend to be poorer and more food insecure compared to male-headed households [62–64].

**Table 4.** Distribution of households based on socio-economic and food security indices.

| Variables | Food Security Indicators | | |
|---|---|---|---|
| **Age** | $\theta_0$ | $\theta_1$ | $\theta_2$ |
| 21–30 | 0.6363 | 0.3475 | 0.2237 |
| 31–40 | 0.4815 | 0.1932 | 0.0969 |
| 41–50 | 0.4211 | 0.1486 | 0.0756 |
| 51–60 | 0.3636 | 0.2222 | 0.0524 |
| 61–70 | 0.1538 | 0.1199 | 0.0998 |
| **Gender** | | | |
| Male | 0.4035 | 0.1466 | 0.0707 |
| Female | 0.4133 | 0.1769 | 0.1003 |
| **Marital status** | | | |
| Married | 0.4681 | 0.1871 | 0.1048 |
| Single | 0.4828 | 0.2037 | 0.1085 |
| Divorced | 0.0000 | 0.0000 | 0.0000 |
| Widow(er) | 0.2857 | 0.0726 | 0.0238 |
| **Educational attainment** | | | |
| Standard (primary school) | 0.2894 | 0.1157 | 0.0657 |
| Matric (high school graduates) | 0.5217 | 0.1986 | 0.0867 |
| Diploma | 0.3158 | 0.1169 | 0.0506 |
| University degree | 0.5217 | 0.2134 | 0.1382 |
| Post graduate | 0.1429 | 0.1374 | 0.1322 |
| **Household size** | | | |
| 1–4 | 0.3396 | 0.1432 | 0.0815 |
| 5–8 | 0.6538 | 0.2205 | 0.0929 |
| 9–12 | 1.0000 | 0.7051 | 0.4971 |
| **Major occupation** | | | |
| Civil servants | 0.4286 | 0.1312 | 0.0465 |
| Entrepreneurs | 0.4902 | 0.2095 | 0.1182 |
| Farmers | 0.4000 | 0.1652 | 0.1017 |
| Unemployed | 0.2500 | 0.0799 | 0.0309 |
| Traditional healers | 0.4286 | 0.2049 | 0.1052 |
| Others | 0.3333 | 0.1578 | 0.0747 |
| **Monthly income** | | | |
| R1000–3000 | 0.3508 | 0.1403 | 0.0710 |
| R3001–5000 | 0.3333 | 0.1534 | 0.0902 |
| R5001–7000 | 0.6666 | 0.2580 | 0.1448 |
| R7001–9000 | 0.4706 | 0.1743 | 0.0899 |

In terms of marital status, single-headed households had the highest (48.28%) incidence of food insecurity, needing 20.37% in per capita food expenditure on a daily basis to be food secure. However, married counterparts recorded 46.81% food insecurity incidence and 18.71% depth, which means the married participants heading the household would require 18.71% of the food

insecurity line (R118.68) per day to be considered as food secured. The current finding suggests the importance of marriage, as both partners might contribute to the food need of the family. Similarly, high rate of food insecurity among households headed by single individuals was observed among rural households in Nigeria [65]. This finding indicates that married household heads have higher likelihood of being food secure than their single household head counterparts in the study area. In the current study, educational attainment of the household heads shows a high (52.17%) food insecurity incidence among degree holders. Similarly, participants with matric qualifications (high school graduates) also had 52.17% food insecurity incidence. However, the food insecurity depth percentage was different, as they would need 19.86% of the food insecurity line of this study to be food secure. There was higher food insecurity among degree holders in rural communities in the North West province of South Africa. This might be due to high level of unemployment in the country or due to location of the participants, which could invariably lead to poverty and food insecurity.

Relative to size, households with 5–8 members had higher (65.38%) incidences of food insecurity compared to smaller households (1–4 members) with 33.96% food insecurity incidence. This shows that a large household size could lead to food insecurity in the study area. A similar trend was also observed in other studies conducted in countries such as Ghana [39] and Kenya [66,67]. In addition, the occupation of the participants showed a food insecurity incidence of 49.02% and 20.95% depth for the entrepreneurs. This indicates a high food insecurity incidence among the entrepreneur categories among the selected communities. The report is also common among other occupational categories such as farmers, civil servants, and traditional healers. This reflects the economic vulnerability level of the household heads that are entrepreneurs, as diverse economic shocks do have adverse effects on their small scale business enterprises and profitability, thereby affecting the households' food base in the study area.

The food insecurity incidence was highest with the participants with monthly income categories R5001–7000, having 66.66% food insecurity incidence and depth of 25.80%, which means that household heads within this monthly income category would need 25.80% of the food insecurity line (R118.68) per day to be food secure. This income category indicates a low income for households in such a category. This corroborates existing literature that low income leads to food insecurity, as such household heads have limited purchasing power [68,69]. On the contrary, households with lower monthly incomes (R1000–3000 and R3001–5000) were more food secure, having 35.08% and 33.33% food insecurity incidence, respectively. This contradicts the *apriori* expectation of this study. However, this might be due to the fact that lower income households adopt proper food insecurity coping mechanisms (e.g., scavenging, skipping of meals, borrowing, buying food in bulk, support from relatives and loved ones) that give them leverage over food insecurity and hence make them more food secure than the higher income households.

*4.3. Factors Influencing Indigenous Plants Inclusion for Households' Food Security*

Binary logistic regression results show that the model fitted the data well, as shown by statistical significance of the chi$^2$ ($p < 0.01$). In addition, the test for multicollinearity among the variables was conducted with variance inflation factor (VIF); the mean VIF of 1.25 was derived in the analysis. Moreover, the high levels of tolerance computed for the variables indicate statistical significance; therefore, the null hypothesis of the study was rejected in favour of the alternative hypothesis. The model used the different parameters for the households, including socioeconomic characteristics, indigenous plant consumption profile, and food security status, which was a binary variable with value 1 if participant was food secured and 0 otherwise. As shown in Table 5, participants' age was positive (0.04335) and significant ($p < 0.10$). This indicates that age of the head of households has the likelihood of influencing the households' food security status. This implies that older-headed households had a higher chance of being food secure. This may be due to the accrued experience with age for such households and that the majority of them were civil servants.

**Table 5.** Factors influencing the inclusion of indigenous plants for food security by households in the study area.

| Variables | Coefficient | Std. Error | Z | P > \| z | Marginal Effects |
|---|---|---|---|---|---|
| Age of the household head | 0.04335 | 0.02277 | 1.90 | 0.057 * | 0.00955 |
| Gender of the household head | 1.02363 | 0.49290 | 2.08 | 0.038 ** | 0.22564 |
| Marital states of the household head | −0.17385 | 0.26320 | −0.66 | 0.509 | −0.03832 |
| Educational attainment of the household head | 0.14116 | 0.03164 | 4.46 | 0.000 *** | 0.02167 |
| Religion of the household head | 1.19710 | 0.65443 | 1.83 | 0.067 * | 0.26387 |
| Households income | 0.05736 | 0.09758 | 0.59 | 0.557 | 0.01264 |
| Households size | −0.62062 | 0.20418 | −3.04 | 0.002 *** | −0.13680 |
| Number of working class | −0.26230 | 0.20791 | −1.26 | 0.207 | −0.05782 |
| Occupation of the household head | 0.31405 | 0.19322 | 1.63 | 0.104 | 0.06922 |
| Inclusion of indigenous plants in diet | 0.00001 | 0.00000 | 3.37 | 0.001 *** | 0.00000 |
| knowledge of indigenous plants by the household head | −0.84305 | 1.07713 | −0.78 | 0.434 | −0.18583 |
| Drought resistant nature of indigenous plants | −0.09914 | 0.45986 | −0.22 | 0.829 | −0.02185 |
| Households food expenditure | −0.49689 | 0.26694 | −1.86 | 0.063 * | −0.07801 |
| Expenditure on indigenous plants | 0.00364 | 0.00236 | 1.54 | 0.123 | 0.00080 |
| Households accessibility to indigenous plants market | 0.77231 | 0.24711 | 3.13 | 0.002 *** | 0.17024 |
| Implementing policies on indigenous plants | 0.66801 | 0.25934 | 2.58 | 0.010 ** | 0.10559 |
| Backyard cultivation of indigenous plants | −0.480941 | 0.72549 | −0.66 | 0.507 | −0.10601 |
| Mainstreaming the indigenous plants into food system | 1.74243 | 0.59993 | 2.90 | 0.004 *** | 0.17696 |
| Constant | 0.498986 | 3.49453 | 0.14 | 0.886 | |
| Observation Number | 133 | | | | |
| LR chi$^2$ (18) | 46.57 | | | | |
| Prob > chi$^2$ | 0.0000 | | | | |
| Pse udo R$^2$ | 0.2644 | | | | |
| Log likelihood | −64.771675 | | | | |

Note: ***, ** and * means 1%, 5%, and 10% levels of significance, respectively.

The current study indicates that the gender of head of the households was positive (1.02363) and significant ($p < 0.05$), suggesting that a male-headed household had a higher probability of being food secure when compared with their female counterparts. This might be due to the fact that more males have higher income generating activities than their female counterparts. Oluwatayo [70] reported that there are more food secure male-headed households when compared to female heads. In addition, the coefficient of the education status of the head of the households was positive (0.14116) and significant at ($p < 0.01$), an indication that the educational level of the head of the households had higher probability of leading to a food secure status in the study area. As previously established in a similar study carried out in Ethiopia, the educational status of the head of the household is essential to improve the food security status of participants [71]. From the current findings, the effect of religion was positive (1.19710) and significant ($p < 0.10$), thereby influencing the probability of being food secure in the study area. The descriptive data indicated 94% Christians, 2% Muslims, and 4% traditionalists. This implies that participants who are Christians have a higher chance of being food secure than their counterparts in other religious beliefs. This could be due to the dominance of Christianity in the selected communities, which might have possibly contributed to a form of religious-based food support within the particular religious community that was lacking in their contemporaries in other religions. However, this could be peculiar to the study area and not generalizable. In addition, the coefficient of household size was negative (−0.62062) and significant at ($p < 0.01$) level of significance. This indicates that the size of the households had a significantly lower probability of influencing the food security status of the participants in the study. This might be due to the lower mean household's size (4 members) recorded in the study. Small-sized household probably translates to lower food demand and hence a food secure household.

On the other hand, the inclusion of indigenous plants in diet by households, captured in dummy form, was positive (0.00001) and significant ($p < 0.01$). The inclusion of indigenous plants in daily diet by households had the probability of making them food secure. It shows that people who are food secure are slightly more likely to include some indigenous foods in their diet than people who are food

insecure. The most likely explanation is that that they live near markets that sell indigenous plants. It is probably the case that people who live near large food markets that sell a diversity of food types are more food secure, regardless of whether those markets sell indigenous or non-indigenous foods. This observation aligns with existing research that the inclusion of indigenous plants in diet could help to reduce food insecurity and strengthen the food system [14,16,20]. Furthermore, the food expenditure (captured in Rands) was negative (−0.49689) and statistically significant ($p < 0.10$). This indicates a negative relationship between the food expenditure of households and their food security status.

In this study, access to indigenous plants market was positive (0.77231) and statistically significant ($p > 0.01$) to the households' food security status. This indicates a positive relationship between market accessibility and food security. It further implies that households that have access to indigenous plants in markets in the study area have a higher likelihood of being food secure than their counterparts without access. The implementation of policies on indigenous plants (captured in its dummy form) had a positive (0.66801) and significant effect on the food security status of households at ($p < 0.05$) level, as expected apriori. Generally, if policies supporting indigenous food plant inclusion in food systems could be implemented, there is a higher likelihood of having a more food secure status among rural communities in South Africa [29,42]. The coefficient of mainstreaming the indigenous plants into the food system dummy variable was also found to be positive (1.74243) and significant ($p < 0.01$). This shows that mainstreaming indigenous plants into a food system could influence the likelihood of being food secure in the study area. This is in line with the assertion by several authors [16,20,40], whereby mainstreaming and integration of indigenous plants for food security could be used to leverage the incorporation of such undervalued varieties to improve access to dietary diversity among the previously disadvantaged populace.

*4.4. Knowledge, Perception, and Identification of Indigenous Plants by Participants*

Table 6 indicates the knowledge level and the perception of indigenous plants for food nutrition security and economic sustainability among the participants. In the study area, 95.49% of the participants were knowledgeable about indigenous plants. In addition, the parents were the source of the knowledge for the majority (40.60%) of the participants. On the other hand, 30.08% learnt about the indigenous plants from their community members. These variables are key to a sustainable food system fortification, as evidence exists that the need for indigenous plants knowledge and exploration is key to a diversified food system [16,20,72].

According to the majority of the participants, indigenous plants are nutritious (69.17%) and healthy (84.21%) for consumption. This is in line with the findings in existing literature [20,72–74]. The studies confirm in various ways that indigenous fruits, vegetables, and grains are nutritious and have diverse health potentials. Likewise, when asked about the possible economic value/potential of indigenous plants, the majority (78.90%) of the participants indicated that the plants have great economic potential, while most (93.23%) of them added their willingness to buy if they found these indigenous plants in the market. In addition, the majority (91.73%) of the participants equally showed that the plants have market potential if explored. This corroborates the previous findings [75–77], which demonstrated that indigenous food plants have available local and international markets as well as the willingness to pay by consumers if these plants are commercially cultivated.

In response to whether the participants acknowledge the potential advantage of the indigenous plants if incorporated into the existing food system, 90.23% of them agreed that the indigenous plants are capable of reducing food insecurity in South Africa. In addition, about 43.61% of the participants indicated that these indigenous plants are sources of nutrients that can mitigate food insecurity, while 37.59% agreed that indigenous plants can be a potential source of nutrition and income (finance) if properly explored. This is in line with the *apriori* expectation as established in many studies [14,15,20], whereby the need to empower the historically underprivileged and marginalized communities through the inclusion of the underutilised indigenous plants in small holder farming holds a panacea for food insecurity in South Africa.

**Table 6.** Participants' knowledge and perception about indigenous plants.

| Variables | Frequency | Percentage | S.D |
|---|---|---|---|
| **Are you knowledgeable about indigenous plants?** | | | |
| Yes | 127 | 95.49 | (0.21) |
| No | 6 | 4.51 | |
| **Source of knowledge** | | | |
| Parents | 54 | 40.60 | |
| Peers | 15 | 11.28 | |
| Community | 40 | 30.08 | (1.26) |
| Traders | 17 | 12.78 | |
| Others | 7 | 5.26 | |
| **Are indigenous plants nutritious?** | | | |
| Yes | 92 | 69.17 | (0.44) |
| No | 41 | 30.83 | |
| **Are Indigenous plants healthy?** | | | |
| Yes | 112 | 84.21 | (0.54) |
| No | 21 | 15.80 | |
| **Do indigenous plants have economic value?** | | | |
| Yes | 105 | 78.90 | (0.41) |
| No | 28 | 21.79 | |
| **Are you willing to pay for indigenous plants?** | | | |
| Yes | 124 | 93.23 | (0.34) |
| No | 9 | 6.77 | |
| **Is there potential market for indigenous plants?** | | | |
| Yes | 122 | 91.73 | (0.23) |
| No | 11 | 8.27 | |
| **Can indigenous plants reduce food security?** | | | |
| Yes | 120 | 90.23 | (0.22) |
| No | 13 | 9.77 | |
| **How can indigenous plants guarantee your food security?** | | | |
| Income/finance generation | 20 | 15.04 | |
| Food provision | 58 | 43.61 | (0.57) |
| Income/finance generation and food provision | 50 | 37.59 | |
| Other means | 5 | 3.76 | |
| **Most indigenous plants are drought, pest, and disease resistant as well as low input required** | | | |
| Yes | 124 | 93.23 | (0.27) |
| No | 9 | 6.77 | |
| **Cultivation of indigenous plants can contribute to environmental sustainability** | | | |
| Yes | 111 | 83.46 | (0.24) |
| No | 22 | 16.54 | |
| **Are there untapped potential in the indigenous plants?** | | | |
| Yes | 126 | 94.74 | (0.26) |
| No | 7 | 5.26 | |
| **Do you plant any indigenous crop?** | | | |
| Yes | 64 | 48.12 | (0.50) |
| No | 69 | 51.88 | |
| **Total** | **133** | **100** | |

The study identified that most of the participants (93.23%) considered indigenous plants as drought, pest, and disease resistant as well as low-agricultural input required plants. In addition, 83.46% of the participants agreed that the cultivation of indigenous plants is capable of sustaining the ecosystem. The majority of the participants (94.74%) indicated that there is untapped potential in the indigenous plants. Indigenous plants may increase food supply, and it is also likely that people who are food insecure have a stronger motivation to grow such plants. In the current study, 51.88% of the participants do not cultivate indigenous plants either for personal consumption or for commercial purpose. This implies that these rural dwellers have knowledge about these indigenous plants but are not cultivating it for consumption or sale. This corroborates existing evidence that cultivation of indigenous plants remains neglected and is often seen as food for the poor, unlike their exotic counterparts, a culture that is mainly attributed to a ripple effect of the apartheid regime of South Africa [15,20].

## 5. Conclusions and Recommendations

The study revealed that 40.6% of the rural households in the selected communities of North West province were food insecure. In addition, age, gender, educational attainment, household size, inclusion of indigenous plants in diet, food expenditure, accessibility to market for indigenous plants, and mainstreaming the indigenous plants into the food system were the significant determinants of household food security status in the study area. Based on the knowledge and the perception of the participants, 95.49% of the participants were knowledgeable about indigenous plants, and 43.61% indicated that these plants are sources of nutrients. In addition, 37.59% agreed that indigenous plants are potential sources of nutrition and income but responders do not cultivate or include them regularly in their diets. Therefore, the study identified some constraints, such as accessibility to market, household size, employment, and food policy. Based on the findings of the study, the following recommendations are essential to enhance the existing food system of South Africa and ensure a more food secured nation:

➢ People in close proximity to markets selling indigenous foods have greater food security. Therefore, greater assistance in getting indigenous foods into markets close to people who are food insecure remains pertinent.

➢ An increased orientation on the importance of family planning and birth control measures is necessary to have a manageable family size that will subsist on the available resources of families in the rural settings of the North West Province.

➢ Unemployment was identified as one of the attributes that leads to food insecurity in the study. More employment schemes and opportunities should be implemented for the increasing South African youths who migrate to urban areas out of frustration from poverty and food insecurity.

➢ Agricultural and food policies at the national level should be aimed at ensuring that the marginalized rural communities are encouraged and trained on the cultivation of indigenous plants to facilitate their inclusion in the food system.

**Author Contributions:** Conceptualization, A.O.O. and A.O.A.; funding acquisition; A.O.A.; investigation; A.O.O.; project administration, A.O.A.; resources, A.O.A.; supervision, A.O.A.; writing original draft, A.O.O.; writing review and editing, A.O.O. and A.O.O. Both authors have read and agreed to the published version of the manuscript.

**Funding:** The research was partially funded by the National Research Foundation (NRG Grant no UID: 109508), Pretoria, South Africa. A.O.A. appreciates the financial support from the Food Security and Safety Niche Area, Faculty of Natural and Agricultural Sciences, North-West University (NWU), Mmabatho, South Africa. The Article Processing Charge (APC) was paid by the North-West University, South Africa.

**Acknowledgments:** A.O.O. appreciates the Postdoctoral fellowship from the North-West University, Mmabatho, South Africa. We thank Peter Tshepiso Ndhlovu and Seleke Christopher Tshwene for assisting with data collection.

**Conflicts of Interest:** The authors declare no conflict of interest. The opinions, conclusions/recommendations herein this study are based on the findings of the authors, therefore, the funders (NRF and NWU) accept no liability whatsoever in this regard.

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
