# Peer review of "Evaluation of Factors Influencing the Inclusion of Indigenous Plants for Food Security among Rural Households in the North West Province of South Africa"

_sustainability, doi:10.3390/su12229562_

Round 1

Reviewer 1 Report

The article needs to add contextual information that would help understand the scope and results of the study:

  • why was the North West province selected as a study area? Because of its poverty and food insecurity? If so, what was known about these before the study?
  • What do people in the North West province eat nowadays, what is their current diet, which foods would be replaced by the indigenous plants? Is diet a problem? Are overweight and obesity a problem in the Province or is food insecurity understood only as shortage of food and the inability of people to eat enough? Or do they eat badly? (South Africa imports processed foods...are these part of the people’s diets in the province?)
  • Most important, which indigenous plants is the article talking about?   This is not clear throughout the text. Is there a definition of what is considered an ‘indigenous crop’ in South Africa?   What did the people interviewed understand under the term ‘indigenous crops’? Which plants for which diet? What is their nutritional value?
  • What is the ecology and farming system of the province? The article mentions the production of maize only…Has maize displaced the indigenous crops?  What size are the farms and do farmers only cultivate maize or other crops, how much for the market and for home consumption?
  • Which policies did the study consider and surveyed participants understood “…as supporting indigenous food plant inclusion in food system…”(line 286)? Is there something wrong with the current policies of the South African government with regard to agriculture?

Given the ambition of the study to contribute to “…the inclusion of indigenous plant-based food policies for a more sustainable food system in South Africa”  (lines 66-67), the article would benefit from adding in the introduction reference to the literature on the linkages between agro-biodiversity, diets, food security and food system (e.g. The Lancet Commission report on Food in the Anthropocene: the EAT–Lancet Commission on healthy diets from sustainable food systems and the studies by Bioversity International). 

More specific comments:

  • line 67-68. The sentence “For the current study, it is hypothesized that the food security status of the participants does not have relationship with heir socio-economic characteristics in the study area” is not clear.
  • Line 226. The sentence is not clear “the report is also common among other occupational categories…traditionalists” (what is meant by ‘traditionalists’?)
  • Line 260 Interesting piece of the findings on religion: merits further explanation.   How does religion affect consumption practices?
  • 296, 1315-317. If 95.48% of the participants were knowledgeable about indigenous plants and 43.61% indicated that these plants are source of nutrients …and 37.59% agreed about their being a potential source of nutrition and income, why is it that -particularly the farming population - do not cultivate them and eat them (statement that you pose in line 329)?   Are you aware of the people’s understanding and own definitions of what is nutritionally good?
  • The conclusions and recommendations: they appear to be very generic.   The authors should first highlight and summarize the constraints identified in their research to the cultivation or purchase of the indigenous plants by the people of the area; then address the specific constraints in policy terms, but with information on the gaps in the current policies or gaps in their implemetation.  

Reviewer 2 Report

The manuscript by Olusola Omotayo and Oldapo Aremu entitled “Evaluation of factors influencing the inclusion of indigenous plants for food security in rural households in the North West province of South Africa” has main aim to detect and analyse the factors that can have influence in inclusion indigenous plants to the diet with purpose to ensure better food security to the residents.

Data were collected among households in 4 different districts with chosen three communities for sampling in each one. Total 188 questionnaires were collected and 133 of them were properly filled.

Authors concluded that around 40% of rural households included were food insecure and that age, gender, education, household size, food expenditure, etc. were determinants of household’s food security status.

Some of the comments/suggestions for the authors:

Row 18 –you state questionnaire is pre-tested and in row 115 you state that you had pilot-study – did You published your data somewhere? Also You don’t have description of questionnaire, only that 31 indigenous plants were included

Row 38 –better to say affected

Row 39 – as with replace with as well as

In introduction part You are mixing terms indigenous food and indigenous plants in same sentences which gives the impression that the only indigenous plant is indigenous food, and what about other, for example, animals?

Also pay attention in the last part of introduction on tenses.

How is possible to assume that food security for family is not in relationship with their socio-economic status?

Row 95 – I’m not sure that is good expression stage 2 – with that we get impression that you had first experimental part during stage 1, then during stage one; and actually at the begging You made decision in which districts and communities you plan to conduct research

Row 107 – what do you mean under WELL-structured?

Row 109 – add were included (after reference 23)

Row 111 – until part with the results it can’t be discerned that the survey was conducted only with the head of the family; here you need to give some description e.g. size of households, etc.

Row 142 – parenthesis is missing

Row 151 – q?? there is in q in equation

You gave table 3, but it would be good to have study population shares for this data as well

Monthly income in table 3 is not clear

Regarding to the monthly income (last part in part 3.2.), maybe is better to present results as income per household member (it’s not the same is there 4 or 6 members having same income)

Explain terminology “matric qualification” row 214

Row 249 – is it possible that this people have higher income?

In the text you have term “dummy” – can you give explanation what do you mean under that term?

From which data you come to conclusion and recommendation about planning family- how does it matter. All four of bulletins in the conclusion are a little too general

In whole manuscript you don’t have any example of those 31 indigenous plants and I assume that this is interesting to the readers.

English proofreading is required

Reviewer 3 Report

Overall, this is a well-presented and well-analysed paper covering an important topic. The most valuable data appears to be that showing how food insecurity is related to factors such as household size, age and education. With regards to indigenous plants, the most useful results are the perceptions of local people that they can help with food security. However, there is a bit of overreach around some of the correlations between indigenous plant consumption and food security. Your data does not show that consuming indigenous plants makes people less food insecure than they otherwise would be and you need to be careful to avoid saying this when it is not justified by the data.

Specific comments are below:

  • Please justify why "it is hypothesized that the food security status of the participants does not have relationship with their socio-economic characteristics in the study area." (line 67-68). I would have thought you would hypothesize that food security is related to socio-economic status. Is there a reason why you have hypothesized the opposite?
  • Table 3 shows that households with low monthly incomes (R1000-3000 and R3001-5000) are actually more food-secure than households with higher incomes. This point requires some discussion, especially as it contradicts the statement that "This corroborates existing literatures that low income leads to food insecurity" (lines 235-236). Perhaps you could reanalyse the data using statistics for monthly income per household member rather than just monthly income.
  • Line 242 states that "the null hypothesis of the study was rejected" but you have not previously stated what the null hypothesis is. 
  • The following sentences need more discussion: "Also, holding other factors constant, a unit increase in households’ inclusion of indigenous plants in their diet will increase the probability of being food secured by 0.0000. This observation corroborates existing research that indigenous plants inclusion in diet could help to reduce food insecurity and strengthen the food system (lines 273-276). Firstly, 0.0000 is zero, so there really is no increase in the probability of being food secured. Secondly, you appear to be confusing correlation with causation. Your data does not actually show that including indigenous food in one's diet increases their level of food security. Rather, it shows that people who are food secure are slightly more likely to include some indigenous foods in their diet than people who are food insecure. The most likely explanation for this appears to be that they live near markets that sell indigenous foods. But it's probably the case that people who live near large food markets that sell a diversity of food types are more food secure anyway, regardless of whether those markets sell indigenous or non-indigenous foods.
  • At line 333, you should note that your data did not show a significant relationship between cultivating indigenous foods and being food secure. This is unsurprising, as while indigenous foods may increase food supply, it is also likely that people who are food insecure have a stronger motivation to grow such plants, so the end result may be that some people growing indigenous plants are food secure while others are food insecure.
  • The conclusion that "there is need for formal education on the importance of indigenous plants as a means of alleviating food insecurity" (line 345) is not really justified by the results. You found that people with higher educational attainment were more likely be food secure, but you did not find any evidence that formal education about indigenous foods helped with food security. On the contrary, you found that most people already knew that indigenous foods could help with food security and that they got this knowledge from their parents and other community members. Why the need for formal education around this?

  • Perhaps you could add a recommendation around improving distribution networks for indigenous foods. I feel that the most promising finding you have unearthed around indigenous foods is that people living close to markets that sell indigenous foods have greater food security. So perhaps what is needed is greater assistance in getting indigenous foods into markets close to people who are food insecure.

Round 2

Reviewer 1 Report

Thank you authors for taking into consideration my comments.   The paper is now fine with me and ready for publication.  There are only two minor points related to the content, suggestions on spelling and one observation, indicated below.  

Content:

  1. The current diet pattern rely on the exotic food varieties such as….it would be useful to know which exotic food varieties. Are they “wheat, rice and potatoes” as mentioned in the National Development Plan?
  2. “Evidence abounds that food production in South Africa is increasing but this is not enough to meet the needs of the increasing population”. A clarification would be useful:  production is increasing of which crops?   Or production of all crops?

English:

Line 35.  Instead of ‘using’  …from? a sustainable food system

  1. These 62 identified gaps need to be filled through an interdisciplinary research approach
  2. do not have (plural) a relationship
  3. can deliver
  4. their food security status, socio-econimics? Do you mean their food security and socio-economic status?
  5. “In addition, the North West Province has a rich biological and ethnic diversity 150 in South Africa”…ethnic referring to population? Or just rich biodiversity?

351  “making them food secured”  …food secure

Observation:  no need to modify content, if you do not think so.

  1. “Food production has to increase geometrically to mitigate this challenge [3, 28)”:  however, be aware that the  National Development Plan does not aim at self-sufficiency  (p.231  The national food security goal should be to maintain a positive trade balance and not to strive for food self-sufficiency in staple foods at all costs.). 

Content:

  1. The current diet pattern rely on the exotic food varieties such as….it would be useful to know which exotic food varieties. Are they “wheat, rice and potatoes” as mentioned in the National Development Plan?
  2. “Evidence abounds that food production in South Africa is increasing but this is not enough to meet the needs of the increasing population”. A clarification would be useful:  production is increasing of which crops?   Or production of all crops?
